# Calf Circumference, a Valuable Tool to Predict Sarcopenia in Older People Hospitalized with Hip Fracture

**DOI:** 10.3390/nu14204255

**Published:** 2022-10-12

**Authors:** Keith Borges, Reyes Artacho, Rosa Jodar-Graus, Esther Molina-Montes, María Dolores Ruiz-López

**Affiliations:** 1Department of Nutrition and Food Science, Faculty of Pharmacy, University of Granada, 18071 Granada, Spain; 2Hospital of Neurotraumatology and Rehabilitation, 18013 Granada, Spain; 3Biomedical Research Center, Institute of Nutrition and Food Technology “José Mataix”, University of Granada, 18071 Granada, Spain; 4CIBER Epidemiología y Salud Pública, CIBERESP, 28029 Madrid, Spain; 5Instituto de Investigación Biosanitaria ibs.GRANADA, 18012 Granada, Spain

**Keywords:** sarcopenia, hip fracture, older people, calf circumference

## Abstract

Sarcopenia is an important risk factor for hip fracture in older people. Nevertheless, this condition is overlooked in clinical practice. This study aimed to explore the factors associated with sarcopenia among older patients hospitalized for hip fracture, to identify a predictive model of sarcopenia based on variables related to this condition, and to evaluate the performance of screening tools in order to choose the most suitable to be adopted in routine care of older people with hip fracture. A cross-sectional study was undertaken with 90 patients (mean age 83.4 ± 7.2 years), by assessing sociodemographic and clinical characteristics, anthropometric measures, such as body mass index (BMI) and calf circumference (CC), the functional status (Barthel Index), the nutritional status (MNA-SF), and the adherence to the Mediterranean Diet (MEDAS). Diagnosis of sarcopenia was established according to the criteria of the European Working Group on Sarcopenia in Older People (EWGSOP2). The analysis of variables associated with sarcopenia was performed using multivariate logistic regression models. Clusters of sarcopenia were explored with heatmaps and predictive risk models were estimated. Sarcopenia was confirmed in 30% of hip fracture patients. Variables with the strongest association with sarcopenia were BMI (OR = 0.79 [0.68–0.91], *p* < 0.05) and CC (OR = 0.64 [0.51–0.81], *p* < 0.01). CC showed a relatively high predictive capacity of sarcopenia (area under the curve: AUC = 0.82). Furthermore, CC could be a valuable tool to predict sarcopenia risk compared with the currently used screening tools, SARC-F and SARC-CalF (AUC, 0.819 vs. 0.734 and 0.576, respectively). More studies are needed to validate these findings in external study populations.

## 1. Introduction

Sarcopenia is a skeletal muscle disease (International Classification of Diseases 10th Revision, ICD-10: M62.5) that involves a progressive and generalized decrease in muscle strength and muscle mass [1]. It is recognized as one of the main risk factors for falls and hip fracture in older adults [2]. A large number of observational studies have demonstrated that there is a high rate of unfavorable clinical outcomes in older adults hospitalized for hip fracture and sarcopenia, such as loss of independence, disability, and mortality [3,4,5,6,7].

In Spain, it is estimated that there are between 40,000 and 45,000 hip fractures in older people per year [8]. This number is expected to continue growing in the next decades given the increasingly aging population. The latest projections from the Spanish National Statistics Institute (INE) indicate that in 2050, 31.4% of the population will be aged over 65 years and 11.6% will be octogenarians [9]. Recent estimates also show that the prevalence of sarcopenia in older hip fracture patients in Spain ranged from 21 to 74% in men and 12 to 68% in women, depending on its definition and diagnosis [10]. Sarcopenia and hip fracture are considered major public health problems in the geriatric population requiring substantial costs for their management [8,11].

There is no single diagnostic criterion for the establishment of sarcopenia. The consensus published by the European Working Group on Sarcopenia in Older People (EWGSOP) is the most widely used one. In its latest update in 2019 (EWGSOP2), the use of screening tests to identify potential cases of sarcopenia was reinforced. Moreover, it was recommended to measure muscle strength by handgrip using dynamometry, and muscle mass mainly by dual-energy X-ray absorptiometry or bioimpedance analysis. However, EWGSOP2 also expanded the possibility of detecting this condition by suggesting other techniques and tools to assess muscle properties, included but not restricted to mid-thigh muscle measurement and ultrasound assessment of muscle mass [1]. However, it is observed that even with these advances and new proposals, this condition is still overlooked in clinical practice.

A simple and valid method to identify sarcopenia in clinical settings is urgently needed. Anthropometric tools are easily obtained as part of regular physical checks of patients. Among them, calf circumference (CC) has been showing to predict nutritional risk [12] and muscle mass in older people [13], disability [14], mortality [15,16], bone mineral density [17], and even sarcopenia in recent studies [18,19,20,21,22].

Considering the relevance of the early identification of sarcopenia for the successful treatment of this disease, the aim of this study was to explore factors associated with sarcopenia, to identify a predictive model of sarcopenia based on variables related to this condition, and to evaluate the performance of screening tools in order choose the most suitable to be adopted in routine care of older people with hip fracture.

## 2. Materials and Methods

### 2.1. Study Population

An observational, cross-sectional study was carried out between March 2020 and March 2021, with patients aged 65 years or older, admitted to the Hospital of Neurotraumatology and Rehabilitation (Granada, Spain) for surgical treatment after suffering a hip fracture due to low impact trauma [23]. The exclusion criteria were patients who presented: (i) cognitive impairment or neurological disease; (ii) advanced diseases affecting nutritional status (according to clinical evaluation); (iii) use of cardiac pacemaker; (iv) prominent leg edema; (v) acute pain that did not allow any assessment; and (vi) verbal refusal to participate in the study or to sign the informed consent form. The protocol for this study was approved by the local ethics Committee of Biomedical Research of the province of Granada; Code number 1750-N-18. Informed consent was obtained from all subjects involved in the study. The study was carried out in accordance with the Helsinki Declaration.

### 2.2. Demographic and Clinical Data

The data collected for the study included: sociodemographic data (age, sex, and living arrangement) and clinical data (comorbidities, number of medications, and length of hospital stay). The impact of comorbidities was evaluated using the original version of the Charlson Comorbidity Index (CCI). CCI consists of 19 items corresponding to diseases, which are weighted to provide a total score of the sum of the different pathologies. It calculates 1-year mortality, rating comorbidity as 0–1 point (no comorbidity), 2 points (low comorbidity), and ≥3 points (high comorbidity) [24]. Polymedication was considered when patients concomitantly consumed more than 5 drugs or dietary supplements/day [25]. The collection of information and patient assessment was performed in the first 24–72 h after admission to the hospital, and always before surgery.

### 2.3. Anthropometric Measurements

Weight (kg) and height (cm) were reported by the patient or accompanying person, and body mass index (BMI) was calculated (kg/m^2^). According to the BMI, patients were classified into underweight (<22 kg/m^2^); normal weight (22–26.9 kg/m^2^); overweight (27–29.9 kg/m^2^); and obese (≥30 kg/m^2^) [26]. CC was measured with a flexible measuring tape (Cescorf, Spain, 1 mm) at the point of the maximal circumference. Patients were in supine position with the knee bent and supported on the bed forming a 90 degree angle. The cut-off point for CC was 31 cm in both males and females since a lower value is related to risk of malnutrition in older adults [12].

### 2.4. Functional Status

Functional status was assessed through the Barthel Index (BI), which considers aspects related to feeding, bathing, dressing, personal hygiene, urinary and fecal continence, toilet use, transferring from chair to bed, ambulation, and ability to climb up and down stairs. The patients were classified according to the points obtained, into total dependent (0–20); severe dependent (21–60); moderate dependent (61–90); mild dependent (91–99); and independent (100) [27].

### 2.5. Nutritional Status

Nutritional status was evaluated by the Mini Nutritional Assessment Short Form (MNA-SF), a gold standard test for malnutrition screening and assessment in the older people including hip fracture [28]. This test consists of 6 questions related to appetite, weight loss, mobility, acute illness, neuropsychological problems, and IMC or CC. It has a maximum score of 14 points and allows the patient’s nutritional status to be classified as: normal nutritional status (12–14); risk of malnutrition (8–11); and malnourished (0–7) [29].

### 2.6. Adherence to the Mediterranean Diet (MedDiet)

The adherence to the MedDiet was evaluated with a questionnaire validated for the Spanish population, the Mediterranean Diet Adherence Screener (MEDAS). MEDAS consists of 14 items that refer to the intake of the most characteristic foods of the MedDiet including: olive oil, fish, nuts, fruits and legumes. To obtain the score, a value of +1 was assigned to each of the items with a positive connotation related to MedDiet and −1 when it had a negative connotation, for example for red meats, commercial bakery, sugary drinks, etc. Based on the obtained score, subjects were categorized into high level of adherence (≥9 points) or low level of adherence to the MedDiet (1–8 points) [30].

### 2.7. Diagnosis of Sarcopenia

The diagnosis of sarcopenia was based on the EWGSOP2 criteria, adapted to the study setting. Probable sarcopenia was considered when low muscle strength was detected, and the diagnosis was confirmed when there was concomitant low muscle strength and muscle mass. Importantly, we could not determine the severity of this condition. Severe sarcopenia is diagnosed based on low performance, for which gait speed is evaluated. However, it was not possible to evaluate this parameter because patients were bedridden. Therefore, in this study, we considered confirmed sarcopenia based on the aforementioned criteria, albeit without assigning the degree of severity [1].

Muscle strength was assessed with a digital hand grip dynamometer (Jamar^®^ PLUS+ dynamometer). Three measurements were made on each hand. In case of impossibility due to pain or immobilization the evaluation was carried out only in one arm. During the evaluation, the bed was slightly raised so that the patient was sitting up; the maximum value was used for the analysis. Low grip strength was defined as values < 27 kg for males and <16 kg for females [1].

Muscle mass was calculated by bioimpedance analysis (impedanciometer BIA 101 ASE, Akern Srl, Italy). The analysis was performed in every patient at baseline, early in the morning. The patients were in supine position, with arms separated forming an angle of approximately 45° with the body, and legs also separated at an angle of approximately 30°. The four electrodes were placed distally, two on the hands and two on the feet on the same side of the body. Resistance and reactance were measured at a low intensity alternating current and a frequency of 50 kHz, according to previously described methodology [31]. From the resistance and reactance, the Appendicular Skeletal Muscle Mass (ASMM) was determined, according to the following equation of Sergi et al. (2015) [32]: ASMM (kg) = −3.965 + (0.222 × RI) + (0.095 × weight) + (1.384 × sex) + (0.064 × Xc).

RI (resistivity index) = height (cm)^2^/resistance (Ω), weight (kg), Xc (reactance) (Ω), sex has values of 0 for females and 1 for males.

Since muscle mass is correlated with body size, we adjusted ASMM for height^2^ (ASMM/height^2^). For the diagnosis of low muscle mass, the cut-off points recommended by EWGSOP2 were applied: ASMM/height^2^: <7 kg/m^2^ in men; <6 kg/m^2^ in women [33].

The following sarcopenia screening tests were used: (a) SARC-F test. It is a simple questionnaire of quick application, which is based on the patient’s own perception of the limitations in relation to: strength, walking assistance, rising from a chair, climbing stairs, and falls. Each of these items are evaluated with 0, 1, or 2 points (no difficulty = 0, some difficulty = 1 and great difficulty or disability = 2) [34]. The maximum score to be achieved is 10 points: suggestive of sarcopenia ≥ 4 points; no signs suggestive of sarcopenia < 4. The Spanish validated version has been used [35]. (b) SARC-CalF test (SARC-F combine with CC). From the SARC-F, 0 points were added for women with CC > 33 cm and men with CC > 34, on the other hand, 10 points were added for women with CC ≤ 33 cm and men with CC ≤ 34. According to the sum of points, patients were classified as suggestive sarcopenic (≥11 points) or non-sarcopenic (0 < 11 points) [36].

## 3. Statistical Analysis

Descriptive statistics were used to summarize the data as means and standard deviation (x ± SD) in the case of continuous variables (for normal distributed data), and absolute and relative frequencies (*n* and %) in the case of categorical variables. To establish comparisons between the groups (sarcopenia vs. non-sarcopenia, or by other variables such as gender), the chi-squared test (categorical variables) and the Student’s *t*-test (continuous variables) were used. Regarding the latter, non-parametric statistical tests (Mann–Whitney U test) were used in the case of non-normal distributions of the variables according to the Kolmogorov–Smirnov test; all key variables were normally distributed. Pearson correlation analysis (assuming normal distributed data) was conducted to establish associations between the study variables. Correlation’s coefficients rho was considered to establish weak (rho > 0.2), moderate (rho > 0.5), or high (rho > 0.8) correlations strengths between the variables. We applied unsupervised hierarchical clustering to elucidate potential clusters or subgroups of patients. All variables were scaled by their means. Clustering was performed by rows, i.e., patients, using Euclidean distances and Ward’s clustering method.

To identify variables associated with a higher likelihood of presenting sarcopenia we used multivariate logistic regression models, considering sarcopenia (yes vs. not, as reference) as the dependent variable, and all the study variables as predictors (independent variables). The selection of these variables in the models was based on the Akaike Information Criterion—AIC. As a measure of association, the odds ratio (OR) and corresponding 95% confidence intervals (95% CI) were estimated.

In addition, logistic regression risk models were used to build prediction models of sarcopenia based on the variables analyzed. We applied stepwise selection methods with backward elimination of predictors from the full predictor model. Those variables associated with sarcopenia (*p* < 0.05) were retained in the model. Then, the predictive capacity of the model was evaluated by means of Receiver Operating Characteristic (ROC) curves and the Area Under the Curve (AUC) estimate.

To evaluate the performance of the screening tools (SARC-F and SARC-CalF) and of CC to predict sarcopenia, we calculated the sensitivity and specificity, positive predictive value (PPV), and negative predictive value (NPV). To determine which measure had the best predictive performance, we compared the values. In addition, the best cut-off point for muscle mass and sarcopenia was determined for the CC. Analyses were performed with R statistical software (version 3.9.0). The significance level was set at α = 0.05 (*p* < 0.05 and 95% confidence level) in all analyses. R packages pheatmap and pROC were used for cluster and ROC curve analyses, and stepAIC for stepwise regression.

## 4. Results

### 4.1. General Characteristics of the Participants

We evaluated 90 patients (80 women and 10 men) with a mean age of 83.4 ± 7.2 years. A total of 68.9% lived with family members. The one-year mortality risk was considered extremely high (98.9%) according to the CCI. The mean number of medications usually consumed at admission was 6.7/day (range 3.1–10.3). Most participants (52.2%) had a severe dependence for performing activities of daily living, according to the BI. The majority of patients (90%) had low muscle strength and 65.6% preserved muscle mass. The number of individuals with sarcopenia diagnosis was 30% (Table 1).

Table 2 shows differences between the non-sarcopenic and sarcopenic patients. There were no significant differences by age, living arrangement, and number of medications and length of hospital stay between the two groups (*p* > 0.05). Weight and BMI were significantly different between the groups (*p* < 0.05). Obese patients were 88% less likely to be sarcopenic (compared to normal weight), according to BMI (95% CI 0.00–0.72). CC was significantly lower in the sarcopenic group than in the non-sarcopenic group (*p* < 0.001). We observed that a higher proportion of sarcopenic patients (85.2%) presented a low CC while most of the patients of the non-sarcopenic group (74.6%) presented normal values of CC. Patients with normal CC were 94% less likely to present sarcopenia than those with low CC (95% CI 0.02–0.19). Patients at risk of malnutrition by MNA-SF (compared to normal status) tended to have 3.3 times higher odds of sarcopenia (95% CI 0.93–16.3). There were no significant differences between the groups regarding BI as continuous measure or its categories (*p* > 0.05), nor were there differences with adherence to the MedDiet assessed by MEDAS (*p* > 0.05). In addition, there were no significant differences by sex regarding BMI, CC and MNA-SF (*p* > 0.05). The study sample was mainly comprised of women (N = 80). Characteristics of women patients are shown in Appendix A.

Differences by CC cut-off point, low CC group vs. normal CC group, are shown in Table 3. There were no significant differences between both groups by sex, living arrangement, number of medications, length of hospital stay, CCI, BI, and MEDAS, or its categories. On the other hand, there were statistically significant differences between the groups with respect to age (*p* = 0.004), weight (*p* < 0.001), BMI (*p* < 0.001), and MNA-SF (*p* < 0.001). Specifically, compared to the low CC group, those at risk of malnutrition according to MNA-SF had 80% (95% CI 2.75–53.2) less likelihood of normal CC, and 94% (95% CI 0.02–0.36) less likelihood if they were malnourished. There were no significant differences by sex regarding weight, BMI, and MNA-SF (*p* < 0.05). Characteristics of women patients according to CC are shown in Appendix A.

### 4.2. Correlation and Clustering Heatmaps Analysis

The cluster heatmap showed groups of variables related to sarcopenia (Figure 1). Two clusters of patients were identified; these clusters corresponded to the presence or absence of sarcopenia. Most sarcopenic patients were grouped in the upper part (in red) and the non-sarcopenic patients in the lower part (royal blue). In addition, every patient was clustered into different subgroups according to the relationship with other variables. The sarcopenic group presented lower values for ASMM and grip strength, followed by weight, BMI, and CC. On the other hand, there was a sarcopenic group of patients who had higher values for SARC-F and SARC-CalF and lower values for MNA-SF and BI, whereas sarcopenic patients with low SARC-F and SARC-CalF seemed to have higher values of the abovementioned screening tests. In the non-sarcopenic group, the pattern found was the opposite, with values above the mean of CC and ASMM both as continuous and categorical variables.

Regarding the correlation analyses between the variables (Figure 2), we observed highly positive correlations between weight, BMI, CC, and ASMM, and weak correlations between height and grip strength with CC. MNA-SF and BI were positively correlated with each other, while these variables were negatively and moderately correlated with SARC-F and SARC-CalF. Both were also positively, though moderately (MNA-SF) or weakly (BI), correlated with CC.

### 4.3. Factors Associated with Sarcopenia: Multivariable Regression Models and AIC Criteria

Table 4 shows variables associated with sarcopenia risk, i.e., the likelihood of presenting sarcopenia, according to multivariate regression models and AIC criteria. The model with the lowest AIC value (Model 1) supported that the variables with the strongest association with sarcopenia were BMI and CC. Regardless of other variables (BMI), a CC ≥ 31 vs. CC < 31 cm significantly decreased the likelihood of sarcopenia by 91.0% (95% CI 0.022–0.295). In addition, regardless of CC, for every unit increase in BMI, the likelihood of sarcopenia decreased significantly by 17.7% (95% CI 0.671–0.983) on average. Other variables (BI and MNA-SF) did not influence Model 1 in a significant manner. Indeed, compared to Model 1, OR estimates did not vary by more than 10% in the other models and the AIC criteria did not increase.

### 4.4. Prediction Models for Sarcopenia

Table 5 shows results of the predictive model (Model 1) of sarcopenia based on a stepwise selection procedure of the variables. Variables that showed significant predictive potential for sarcopenia were BMI in kg/m^2^ (*p* = 0.04) and CC in categories (<31 cm and ≥31 cm) (*p* < 0.001).

### 4.5. Cut-off Value of CC to Predict Sarcopenia and ASMM

ROC analysis was performed to determine the ability of the aforentioned prediction Model 1 (based on CC) to appropriately classify the patients as sarcopenic or non-sarcopenic. The AUC using this model was 0.824 (95% CI 0.746–0.853) (Figure 3a) with a specificity of 74.6% and sensitivity of 85.2%, and NPV and PPV of 0.92, 0.58, respectively. The optimal cutoff value for predicting sarcopenia using CC was given at 31 cm. Furthermore, the classification performance of CC into low and normal ASMM was relatively high (Figure 3b), with a value of AUC reaching 0.794 (95% CI 0.774–0.826), a specificity of 77% and a sensitivity of 74.6%.

### 4.6. Performance of Screening Tools (SARC-F and SARC-CalF) and CC in the Detection of Sarcopenia Risk

Table 6 shows CC performance to detect sarcopenia risk compared with the currently used screening tools, SARC-F and SARC-CalF. The model with CC had a higher AUC (0.824, 95% CI 0.746–0.852) than the other screening tools, namely SARC-F (AUC = 0.553, 95% CI 0.286–0.889) and SARC-CalF (AUC = 0.597, 95% CI 0.413–0.852). Thus, the CC model had a better classification accuracy according to these analyses.

## 5. Discussion

In the present study, 30% of hip fracture patients had sarcopenia. Sarcopenia was significantly associated with BMI (underweight), MNA-SF (risk or malnutrition), and especially with low CC. These relationships were corroborated in cluster heatmap and multivariate regression analysis. The predictive performance of BMI and CC for sarcopenia detection was relatively high, the latter being the most relevant in the predictive model. Additionally, the cut-off point of CC to predict sarcopenia in this model was set at 31 cm. The performance of CC compared with currently used screening tests for sarcopenia had higher sensitivity and specificity. This anthropometric measure also proved to be a valid predictor of the ASMM.

The percentage of sarcopenic patients in our study is within the range of 11–76.9% found by other authors who have investigated older patients with hip fracture [20]. For instance, in Spain, Cervera-Días et al. identified the highest percentage of sarcopenia among 186 patients hospitalized for hip fracture, 76.9% (mean age 86.2 years) [37]. On the other hand, in the study of Sánchez-Castellano et al., the sarcopenia prevalence among 150 patients (mean age 87.6 years) was estimated to be between 11.5 and 34.9% according to different equations used to estimate muscle mass (bioimpedance, using two different cut-off points, Janssen and Masanés) [10]. González-Montalvo et al. detected 17.1% sarcopenics among 479 acute hip fracture patients (mean age 85.3 years) [19].

It is worthy of note that the prevalence of sarcopenia in hip fracture patients varies widely, depending on populations, diagnostic tools, and the definition of sarcopenia. In our study, in agreement with previous studies [38,39], poorer nutritional status (by MNA-SF) was closely associated with sarcopenia. This relation was expected, since malnutrition, sarcopenia, and hip fracture are correlated with each other, and are commonly occurring conditions in older people populations [40]. They appear clinically through a combination of decreased body weight and nutrients intake, along with a decrease in muscle mass and bone mineral density. Moreover, malnutrition is one of the key risk factors of sarcopenia [41].

Regarding CC, it is important to highlight that we found a strong relationship between this measure and sarcopenia. Older patients with hip fracture and small CC have higher sarcopenia risk, while larger calf muscles may be protective against sarcopenia. This is a relevant finding because this measurement is cheap, simple and non-invasive, and could be easily implemented in clinical guidelines for the management of sarcopenia. Furthermore, several authors have investigated the relationship between CC and muscle mass in sarcopenic patients from different populations. In community settings, Santos et al. evaluated 15,293 adults from the NHANES 1999–2006 cohort and showed relatively high correlations between CC and ASMM measured by dual-energy X-ray absorptiometry (DXA) (r = 0.79 for men and 0.74 for women, respectively) [38]. Similarly, another study involving 213 individuals (55–75 years old) living in the community showed significant positive and moderate correlations between CC and ASMM measured by BIA (r = 0.57 and 0.60 for women and men, respectively; *p* = 0.0001) [19]. This relationship was corroborated in The Korean Frailty and Aging Cohort Study (N = 657 patients, mean age 76.2 ± 4 years), where CC and AASM were positively correlated with each other (r = 0.55; *p* < 0.001 for women and men) [22]. However, in hospital settings, few studies have tested this relationship.

Endo et al. performed a ROC analysis to diagnose sarcopenia by CC in 525 chronic liver disease hospitalized patients in Japan and concluded that it seemed a useful and simple surrogate tool for screening sarcopenia in these type of patients (AUC of 0.91 for men and 0.89 for women; respectively, optimal cut-off values of CC: 32.6 cm for men (sensitivity, 83.7%; specificity, 84.7%) and 32.1 cm for women (sensitivity, 85.1%; specificity, 81.3%))]. In this study, sarcopenia was defined according to the Japan Society of Hepatology [18]. Likewise, Inoue et al. tested the ability of CC to predict sarcopenia in 256 stroke patients considering the latest update of the Asian Working Group for Sarcopenia criteria, published in 2020 (AWGS2) [42,43]. The CC cut-off points found in these studies were different for men and women, which differs from our results that showed the same CC cut-off values for both sexes. Moreover, in our study, when restricting analyses to women only, the estimates remained the same. This is further supported by the fact that sex had no influence in our models. To facilitate earlier identification of people at risk for sarcopenia, the EWGSOP2 [1] and AWGS2 [43] have proposed diagnostic algorithms starting with screening tests, SARC-F and SARC-CalF. In addition, AWGS2 also included CC as screening test, considering CC < 34 cm in men and <33 cm in women as cut off value [43]; thus, sex-specific cut-off values for CC were given by these criteria, too. As aforementioned, there might be variations in the prevalence, risk factors, and clinical features of sarcopenia according to patient population characteristics, the study sample, and the geographic origin.

When comparing the CC with the commonly used screening tests, we observed that CC presented higher sensitivity and specificity to predict sarcopenia than SARC-F and SARC-CalF (AUC, 0.819 vs. 0.734 vs. 0.576). In line with our results, Chen et al. showed that CC had a higher accuracy compared to SARC-CalF, and SARC-F in older assisted living subjects (AUC = 0.819 vs. 0.734 vs. 0.576), as well as a higher sensitivity and specificity for sarcopenia (N = 236 patients; age: 78.7 ± 8.6 years in men and 81.1 ± 6.8 years in women). However, regarding screening tools our findings were different. The other authors reported a sensitivity/specificity for SARC-CalF and SARC-F of 38.0%/80.0% and 10.9%/91.8%, whereas we estimated a different trend with higher sensitivity but lower specificity values (85.1%/41.2% vs. 88.9%/28.6%) [44]. Again, these somewhat inconsistent findings might be attributed to the characteristics of the participants and the low sample size, since both could influence the estimates.

Finally, it is important to note that in our study, we did not find any significant relationship between BI, CCI, MEDAS, and sarcopenia. While this is an unexpected finding, we believe that these results can be explained by the fact that the study sample comprised mostly older patients with high dependency and mortality risk. Silva et al. (2018) in a systematic review and meta-analysis concluded that adherence to the MedDiet could be protective for frailty and functional disability, but not for sarcopenia [45]. By contrast, other studies have found significant associations between these variables. For instance, Kamijo et al. evaluated older adults on peritoneal dialysis and found inverse an association between high values of BI and CCI and sarcopenia (N = 105; 67  ±  13.5 years old, 73.3% men, 26.7% women) [46]. Hashemi et al. tested whether adherence to a particular dietary pattern was associated with sarcopenia among the older people and found that the MedDiet was associated with a lower odds of sarcopenia [47]. Moreover, Tan et al. found a positive association between sarcopenia and a higher CCI and polymedication [48]. However, while these studies support that some factors are associated with sarcopenia, further studies are needed to more consistently determine factors associated with this condition.

## 6. Strengths and Limitations

This study has some limitations to note. First, the sample size was probably not large enough to observe significant associations between the study variables and sarcopenia. Second, it was unfeasible to measure weight and height of the patients, even though previous studies have shown high inter/intraclass correlation coefficients for self-reported and measured values of weight and height in older adults [49]. Despite this limitation, we used standard protocols to measure CC in this study. Third, while the AUC of CC to predict sarcopenia was high, this estimate lacks external validation. Therefore, it is likely that this estimate and its precision are overestimated. Fourth, there might be collinearity between CC and MNA-SF in the models since we chose to use CC as part of this screening tool (and not BMI).

There are also some important strengths to highlight. To our knowledge, this is the first study in Europe to evaluate the aforementioned performance of CC to predict sarcopenia in older patients hospitalized and having suffered a hip fracture. All potential variables related with sarcopenia were collected from these patients by using valid questionnaires, face-to-face interviews and physical measures of body composition and anthropometry (except weight and height). Our results suggest, for the first time, that CC is a good screening tool to predict sarcopenic individuals among hospitalized older adult with hip fracture. In addition, we used comprehensive statistical methods and several approaches to corroborate our findings. For instance, in cluster analyses, we could verify that the characteristics of sarcopenic patients differ from those of non-sarcopenic patients and that a low CC was present in almost all sarcopenic patients with hip fracture. Our findings further reinforce the importance of including CC as a simple, rapid, noninvasive measurement to be included in the comprehensive evaluation of all older patients admitted with hip fracture. This may initiate awareness of sarcopenia and may help begin to design more specific nutritional interventions and treatment protocols for the improvement of post-surgical rehabilitation and prevention disability, as well as mortality, in these patients.

## 7. Conclusions

In conclusion, the prevalence of sarcopenia in older adults with hip fractures was 30% and the main factors associated with this condition were CC and BMI. CC proved to be a potential valuable tool to predict sarcopenia, with a relatively high sensitivity and specificity to identify sarcopenia among these patients compared to other screening tools (SARC-F and SARC-CalF). Its incorporation into clinical practice should be evaluated and validated in large and external study populations. Thus, further studies are necessary to confirm these findings, i.e., to demonstrate the validity of CC for sarcopenia prediction and its association with sarcopenia and related factors in patients with hip fracture.

## Figures and Tables

**Figure 1 nutrients-14-04255-f001:**
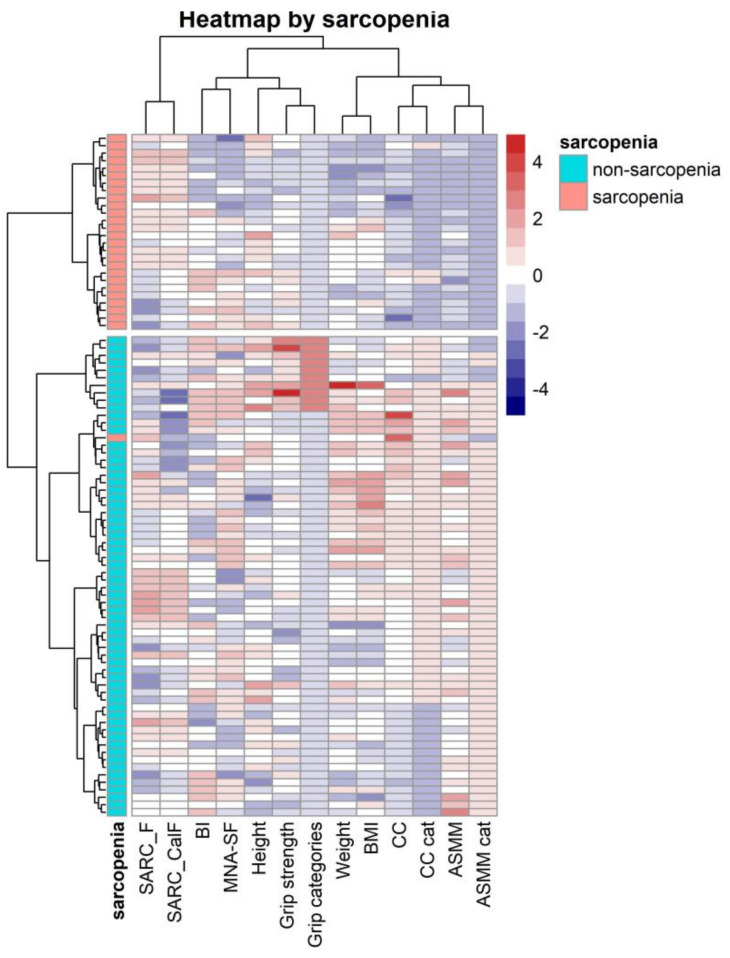
Clustering heatmap illustrating the potential clusters or subgroups of patients, using relevant variables of sarcopenia diagnosis and screening. ASMM (Appendicular Skeletal Muscle Mass); BI (Barthel Index); BMI (Body Mass Index); CC (Calf Circumference); CCI (Charlson Comorbidity Index); MEDAS (Mediterranean Diet Adherence Screener); MNA-SF (Mini Nutritional Assessment-Short Form). All variables were scaled by their means. Clustering was performed by rows, i.e., patients, using Euclidean distances and Ward’s clustering method. Color patterns followed the intensity of this relationship. The redder reflects that this pattern is above average and the bluer means the opposite.

**Figure 2 nutrients-14-04255-f002:**
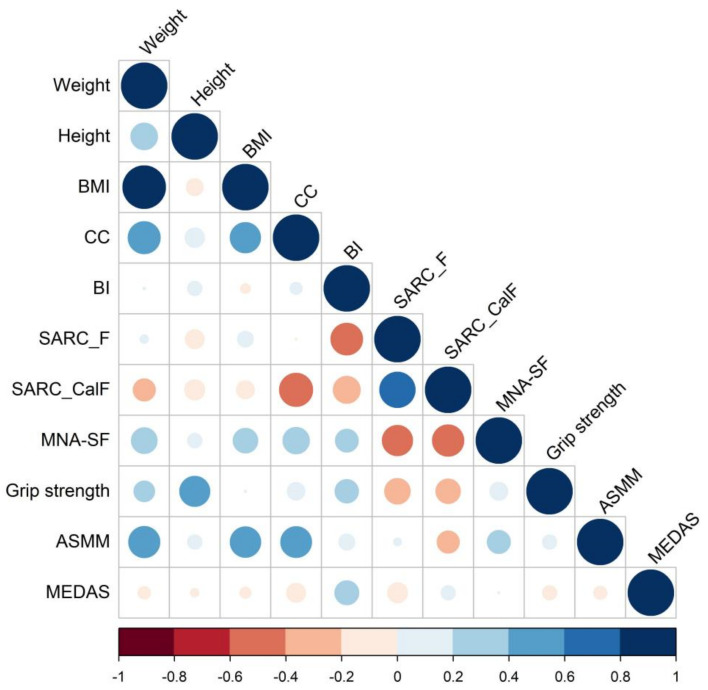
Pearson correlation analysis matrix. ASMM (Appendicular Skeletal Muscle Mass); BI (Barthel Index); BMI (Body Mass Index); CC (Calf Circumference); CCI (Charlson Comorbidity Index); MEDAS (Mediterranean Diet Adherence Screener); MNA-SF (Mini Nutritional Assessment-Short Form). Blue and red colors indicate positive and negative correlations, respectively.

**Figure 3 nutrients-14-04255-f003:**
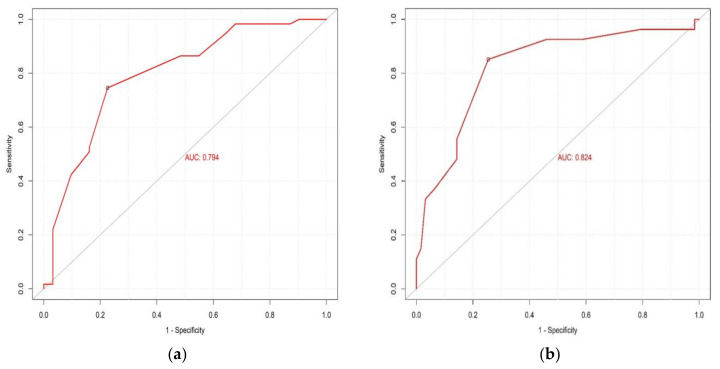
(**a**) Discriminative ability of CC to predict sarcopenia (CC~Sarcopenia). (**b**) Discriminative ability of CC to predict ASMM (CC~ASMM). ASMM (Appendicular Skeletal Muscle Mass); AUC (Area Under the Curve); CC (Calf Circumference); ROC (Receiver Operating Characteristic). Values are indicated in red color. The black circle refers to the CC cut point (set at 31 cm).

**Table 1 nutrients-14-04255-t001:** General characteristics of the study sample (90 older patients with hip fracture).

	*n* = 90
**Age (years)**	83.4 (±7.2)
Sex	
Female	80 (88.9%)
Male	10 (11.1%)
**Living arrangement**	
Other family members	62 (68.9%)
Nursing home residents	10 (11.1%)
Alone	9 (10.0%)
Partner	9 (10.0%)
**CCI**	
High comorbidity	89 (98.9%)
Low comorbidity	1 (1.11%)
No comorbidity	0 (0.0%)
**Mdications/day**	6.7 (±3.56)
**Length of hospital stay (days)**	10.9 (±7.38)
**Weight (kg)**	68.2 (±11.5)
**Height (m)**	1.59 (±0.06)
**BMI (kg/m^2^)**	26.9 (±4.31)
Normal weight	38 (42.2%)
Underweight	11 (12.2%)
Overweight	24 (26.7%)
Obese	17 (18.9%)
Normal weight	38 (42.2%)
**CC (cm)**	31.1 (±3.33)
**CC categories**	
Low	39 (43.3%)
Normal	51 (56.7%)
**BI**	58.8 (±27.4)
**BI categories**	
Independent	3 (3.33%)
Mild dependent	17 (18.9%)
Moderate dependent	16 (17.8%)
Severe dependent	47 (52.2%)
Total dependent	7 (7.78%)
**MNA-SF**	9.94 (±2.63)
**MNA-SF categories**	
Normal	26 (28.9%)
At risk of nutrition	45 (50.0%)
Malnourished	19 (21.1%)
**MEDAS**	0.70 (±0.46)
**MEDAS categories**	
Low	18 (20.0%)
High	72 (80.0%)
**Grip Strength (kg)**	12.1 (±6.80)
**Grip Strength categories**	
Low	81 (90.0%)
Normal	9 (10.0%)
**ASMM/height**^2^ (kg/m^2^)	6.80 (±1.35)
**ASMM/height** ^2^ **categories**	
Low	31 (34.4%)
Normal	59 (65.6%)
**Probable sarcopenia**	81 (90.0%)
**Sarcopenia**	27 (30.0%)

ASMM (Appendicular Skeletal Muscle Mass); BI (Barthel Index); BMI (Body Mass Index); CC (Calf Circumference); CCI (Charlson Comorbidity Index); MEDAS (Mediterranean Diet Adherence Screener); MNA-SF (Mini Nutritional Assessment-Short Form). Values are means (± SD) assuming normal distributed data or frequencies (%).

**Table 2 nutrients-14-04255-t002:** Characteristics of older patients hospitalized for hip fractures according to the presence or absence of sarcopenia.

	Non-Sarcopenic Group *n =* 63	Sarcopenic Group *n* = 27	OR [95% CI]	*p*
**Age (years)**	83.0 (±7.27)	84.1 (±6.97)	1.02 [0.96–1.09]	0.503
**Sex**				
Female	55 (87.3%)	25 (92.6%)	Ref.	
Male	8 (12.7%)	2 (7.41%)	0.58 [0.08–2.60]	0.506
**Living arrangement**				
Other family members	42 (66.7%)	20 (74.1%)	Ref.	
Nursing home residents	7 (11.1%)	3 (11.1%)	0.92 [0.17–3.82]	0.916
Alone	8 (12.7%)	1 (3.70%)	0.30 [0.01–1.83]	0.219
Partner	6 (9.52%)	3 (11.1%)	1.07 [0.20–4.67]	0.930
**CCI**				
High comorbidity	62 (98.4%)	27 (100%)	Ref.	
Low comorbidity	1 (1.59%)	0 (0.00%)		
No comorbidity	0 (0.0%)	0 (0.00%)	ND	
**Nº of medications/day**	6.78 (±3.62)	6.52 (±3.49)	0.98 [0.86–1.11]	0.751
**Length hospital stay (days)**	10.9 (±7.00)	10.9 (±8.33)	1.00 [0.94–1.06]	0.990
**Weight (kg)**	70.9 (±11.5)	62.0 (±8.87)	0.92 [0.87–0.97]	0.001
**Height (m)**	1.59 (±0.06)	1.59 (±0.06)	0.23 [0.00–366]	0.694
**BMI (kg/m^2^)**	27.9 (±4.38)	24.5 (±3.13)	0.79 [0.68–0.91]	0.001
**BMI categories**				
Normal weight	24 (38.1%)	14 (51.9%)	Ref.	
Underweight	4 (6.35%)	7 (25.9%)	2.89 [0.72–13.3]	0.136
Overweight	19 (30.2%)	5 (18.5%)	0.46 [0.13–1.47]	0.198
Obese	16 (25.4%)	1 (3.70%)	0.12 [0.00–0.72]	0.016
**CC (cm)**	32.0 (±2.79)	28.9 (±3.49)	0.64 [0.51–0.81]	<0.001
**CC categories**				
Low	16 (25.4%)	23 (85.2%)	Ref.	
Normal	47 (74.6%)	4 (14.8%)	0.06 [0.02–0.19]	<0.001
**BI**	61.2 (±27.8)	53.3 (±26.2)	0.99 [0.97–1.01]	0.213
**BI categories**				
Independent	3 (4.76%)	0 (0.00%)	Ref.	
Mild dependent	12 (19.0%)	5 (18.5%)	ND	
Moderate dependent	13 (20.6%)	3 (11.1%)	ND	
Severe dependent	29 (46.0%)	18 (66.7%)	ND	
Total dependent	6 (9.52%)	1 (3.70%)	ND	
**MNA-SF**	10.4 (±2.51)	8.85 (±2.63)	0.79 [0.66–0.95]	0.012
**MNA-SF categories**				
Normal	23 (36.5%)	3 (11.1%)	Ref.	
At risk of nutrition	31 (49.2%)	14 (51.9%)	3.30 [0.93–16.3]	0.067
Malnourished	9 (14.3%)	10 (37.0%)	7.84 [1.87–43.9]	0.004
**MEDAS**	0.71 (±0.46)	0.67 (±0.48)	0.80 [0.30–2.11]	0.652
**MEDAS categories**				
Low	13 (20.6%)	5 (18.5%)	Ref.	
High	50 (79.4%)	22 (81.5%)	1.13 [0.37–3.96]	0.841

ASMM (Appendicular Skeletal Muscle Mass); BI (Barthel Index); BMI (Body Mass Index); CC (Calf Circumference); CCI (Charlson Comorbidity Index); MEDAS (Mediterranean Diet Adherence Screener); MNA-SF (Mini Nutritional Assessment-Short Form). ND (not determined). Some estimates could be not determined due to low numbers in strata. Values are means (± SD) assuming normal distributed data or frequencies (%). OR (Odds ratio), CI = 95% (Confidence Interval 95%) [lower CI–upper CI] and *p*-value (*p*).

**Table 3 nutrients-14-04255-t003:** Characteristics of older patients hospitalized for hip fractures according to CC.

	Low CC Group	Normal CC Group	OR [95% CI]	*p*
	*n =* 39	*n =* 51		
**Age (years)**	86.0 (±6.90)	81.3 (±6.75)	0.90 [0.84–0.97]	0.004
**Sex**				
Female	36 (92.3%)	44 (86.3%)	Ref.	
Male	3 (7.69%)	7 (13.7%)	1.85 [0.46–9.59]	0.396
**Living arrangement**				
Other family members	28 (71.8%)	34 (66.7%)	Ref.	
Nursing home residents	5 (12.8%)	5 (9.80%)	0.83 [0.20–3.36]	0.785
Alone	2 (5.13%)	7 (13.7%)	2.71 [0.58–21.3]	0.218
Partner	4 (10.3%)	5 (9.80%)	1.02 [0.24–4.68]	0.976
**CCI**				
High comorbidity	39 (100%)	50 (98.0%)	Ref.	
Low comorbidity	0 (0.00%)	1 (1.96%)	ND	
**Nº of medications/day**	6.74 (±3.09)	6.67 (±3.91)	0.99 [0.88–1.12]	0.919
**Length hospital stay (days)**	10.2 (±7.30)	11.5 (±7.46)	1.03 [0.97–1.09]	0.412
**Weight (kg)**	62.5 (±8.36)	72.6 (±11.8)	1.11 [1.05–1.18]	<0.001
**Height (cm)**	158 (±6.75)	160 (±6.57)	1.07 [1.00-1.14]	0.047
**BMI (kg/m^2^)**	25.0 (±3.04)	28.3 (±4.63)	1.25 [1.09–1.42]	0.001
**BMI categories**				
Normal weight	23 (59.0%)	15 (29.4%)	Ref.	
Underweight	7 (17.9%)	4 (7.84%)	0.89 [0.19–3.59]	0.871
Overweight	7 (17.9%)	17 (33.3%)	3.61 [1.23–11.5]	0.019
Obese	2 (5.13%)	15 (29.4%)	10.4 [2.44–80.0]	0.001
**BI**	55.4 (±27.3)	61.5 (±27.4)	1.01 [0.99–1.02]	0.295
**BI categories**				
Independent	1 (2.56%)	2 (3.92%)	Ref.	
Mild dependent	7 (17.9%)	10 (19.6%)	0.76 [0.02–11.2]	0.849
Moderate dependent	4 (10.3%)	12 (23.5%)	1.53 [0.04–24.2]	0.779
Severe dependent	23 (59.0%)	24 (47.1%)	0.56 [0.02–7.35]	0.663
Total dependent	4 (10.3%)	3 (5.88%)	0.43 [0.01–7.81]	0.583
**MNA-SF**	8.85 (±2.38)	10.8 (±2.52)	1.37 [1.13–1.65]	0.001
**MNA-SF categories**				
Normal	4 (10.3%)	22 (43.1%)	Ref.	
At risk of nutrition	22 (56.4%)	23 (45.1%)	0.20 [0.05–0.63]	0.005
Malnourished	13 (33.3%)	6 (11.8%)	0.09 [0.02–0.36]	<0.001
**MEDAS**	0.74 (±0.44)	0.67 (±0.48)	0.69 [0.27–1.74]	0.431
**MEDAS categories**				
Low	6 (15.4%)	12 (23.5%)	Ref.	
High	33 (84.6%)	39 (76.5%)	0.60 [0.19–1.75]	0.356
**Grip Strength (kg)**	10.4 (±4.08)	13.3 (±8.11)	1.08 [1.00–1.18]	0.054
**Grip categories**				
Low	38 (97.4%)	43 (84.3%)	Ref.	
Normal	1 (2.56%)	8 (15.7%)	6.22 [1.04–162]	0.044
**ASMM/height^2^ (kg/m^2^)**	6.16 (±1.25)	7.30 (±1.21)	2.27 [1.46–3.53]	<0.001
**ASMM/height^2^ categories**				
Low	24 (61.5%)	7 (13.7%)	Ref.	
Normal	15 (38.5%)	44 (86.3%)	9.62 [3.58–29.0]	<0.001
**Sarcopenia diagnosis**				
Non-sarcopenic	16 (41.0%)	47 (92.2%)	Ref.	
Sarcopenic	23 (59.0%)	4 (7.84%)	0.06 [0.02–0.19]	<0.001

ASMM (Appendicular Skeletal Muscle Mass); BI (Barthel Index); BMI (Body Mass Index); CC (Calf Circumference); CCI (Charlson Comorbidity Index); MEDAS (Mediterranean Diet Adherence Screener); MNA-SF (Mini Nutritional Assessment-Short Form). ND (not determined). Some estimates could be not determined due to low numbers in strata. Values are means (± SD) assuming normal distributed data, or frequencies (%). OR (Odds Ratio), CI=95% (Confidence Interval) [lower CI–upper CI] and *p*-value (*p*).

**Table 4 nutrients-14-04255-t004:** Factors associated with sarcopenia according to multivariable logistic regression analysis.

	Model 1			Model 2			Model 3			Model 4			Model 5		
	OR	Lower CI	Upper CI	OR	Lower CI	Upper CI	OR	Lower CI	Upper CI	OR	Lower CI	Upper CI	OR	Lower CI	Upper CI
**BMI (per 1 unit)**	0.823 **	0.671	0.983	0.829 **	0.682	0.982	0.823 **	0.671	0.947	0.837 *	0.688	0.995	0.845 *	0.694	1.005
**CC < 31 cm (Ref.)**	1.00			1.00		1.00			1.00			1.00			
CC ≥ 31 cm	0.090 ***	0.022	0.295	0.085 ***	0.022	0.272	0.086 ***	0.021	0.279	0.086 ***	0.021	0.282	0.091 ***	0.022	0.304
**BI (per 1 unit)**				0.991	0.970	1.012									
**BI categories**															
BI Mild dependent Ref.				1.00			1.00								
BI Moderate dependent							1.443	0.192	10.405						
BI Severe dependent							1.821	0.462	8.298						
**MNA-SF (per 1 unit)**										0.957	0.752	1.219			
**MNA-SF categories**															
MNA-SF normal (Ref.)													1.00		
MNA-SF at risk													1.349	0.278	7.492
MNA-malnourished													1.976	0.323	13.232
Log likelihood		−38.075			−37.716			−37.691			−38.009			−37.782	
Akaike Inf. Criteria		82.149			83.431			85.382			84.018			85.565	

ASMM (Appendicular Skeletal Muscle Mass); BI (Barthel Index); BMI (Body Mass Index); CC (Calf Circumference); CCI (Charlson Comorbidity Index); MEDAS (Mediterranean Diet Adherence Screener); MNA-SF (Mini Nutritional Assessment-Short Form). OR (Odds Ratio), CI = 95% (Confidence Intervals). Model 1: adjusted for BMI (continuous; kg/m^2^) and CC (<31 cm, ≥31 cm); Model 2: Model 1 also adjusted for BI; Model 3: Model 1 also adjusted for BI categories; Model 4: Model 1 also adjusted for MNA-SF index; Model 5: Model 1 also adjusted for MNA-SF categories. *p* < 0.05 was statistically significant: * *p* < 0.1; ** *p* < 0.05; *** *p* < 0.01.

**Table 5 nutrients-14-04255-t005:** Statistical results of the forward stepwise regression prediction model (Model 1) for sarcopenia.

Variable	Parameter Estimate	Standard Error	*z*	*p*
**Intercept**	2.516	2.391	1.052	0.292
**BMI (kg/m^2^)**	−0.185	0.091	−2.037	0.04
**CC < 31 cm**	2.521	0.630	4.000	<0.001

BMI (Body Mass Index); CC (Calf Circumference); *p* value (*p*) and z score (*z*).

**Table 6 nutrients-14-04255-t006:** Performance of screening tools (SARC-F and SARC-CalF) and CC in the detection of sarcopenia risk.

Variables	Sensitivity (%)	Specificity (%)	NPV	PPV	Younden’s Index	AUC
**SARC-F**	88.9	28.6	0.86	0.35	1.17	0.553
**SARC-CalF**	85.1	41.2	0.86	0.38	1.26	0.597
**CC**	85.2	74.6	0.92	0.58	1.59	0.824

AUC (Area Under the Curve); BMI (Body Mass Index); CC (Calf Circumference); NPV (Negative Predictive Value); PPV (Positive Predictive Value).

## Data Availability

Data are available upon reasonable request.

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
