# Peer review of "Calf Circumference, a Valuable Tool to Predict Sarcopenia in Older People Hospitalized with Hip Fracture"

_nutrients, 2022, doi:10.3390/nu14204255_

Round 1

Reviewer 1 Report

As the authors have identified, sarcopenia is a common condition among older hip fracture patients--the topic will be of interest to Nutrients readers. An easy to implement screening measure for sarcopenia, such as the calf circumference measure described in the manuscript, could be beneficial to clinicians and patients.

The manuscript explains the basis for the study and provides detailed information on the methods, analyses and results. Further, the authors provide a good discussion of how their study results compare to the findings of others and identify strengths and limitations of their research.

My only comment is that the manuscript would benefit from minor English review. In some cases words are missing or the verb tense is incorrect.

Author Response

Thank you very much for your comments.

The text has been revised by a native English speaker to improve the style and language

Reviewer 2 Report

Excellent manuscript on surrogate marker of sarcopenia for older patients admitted with Hip Fracture namely calf circumference.

Good introduction and explanation of problem of sarcopenia.

extensive description in Methods and Materials describing the frail study population, demographic , clinical data, anthropometric measurements, functional status , nutritional status , adherence to Mediterranean diet and diagnosis of sarcopenia.

Authors have described and articulated the problem of measuring and assessing sarcopenia. Hence the necessary detailed variables in the results section

Only concern is low number sad highly skewed female %.

Good explanation of heat maps of the differing indices measures.

Agree with discussion and conclusion

Author Response

Thank you very much for your comments.

The test has been revised by a native English speaker to improve the style and language.

Also, regarding the reviewer´s comments on the distribution of women and men in the study sample, we would like to mention that we included before supplementary tables to show characteristics of the study sample restricted to women (line 240-242). We also reported that there were no significant changes between men and women concerning key variables (line 239-240).

In this new version, we have added further that the results remained similar when analyses were restricted to women, and that sex did not influence the estimates in any model (line 420-423).